# CLIP2LE: A LABEL ENHANCEMENT FAIR REPRESENTATION METHOD VIA CLIP

## ABSTRACT

Label enhancement is a novel label shift strategy that aims to integrate the feature space with the logical label space to obtain a high-quality label distribution. This label distribution can serve as a soft target for algorithmic learning, akin to label smoothing, thereby enhancing the performance of various learning paradigms including multi-label learning, single positive label learning, and partial-label learning. However, limited by dataset type and annotation inaccuracy, the same label enhancement algorithm on different datasets struggles to achieve consistent performance, for reasons derived from the following two insights: 1) Differential Contribution of Feature Space and Logical Label Space: The feature space and logical label space of different datasets contribute differently to generating an accurate label distribution; 2) Presence of Noise and Incorrect Labels: Some datasets contain noise and inaccurately labeled samples, leading to divergent outputs for similar inputs. To address these challenges, we propose leveraging CLIP (Contrastive Language-Image Pre-training) as a foundational strategy, treating the feature space and the logical label space as two distinct modalities. By recoding these modalities before applying the label enhancement algorithm, we aim to achieve a fair and robust representation. In addition, we further explained the reasonableness of our motives in the discussion session. Extensive experimental results demonstrate the effectiveness of our approach to help existing label enhancement algorithms improve their performance on several benchmarks.

## 1 INTRODUCTION

Currently, improving the performance of multi-label learning algorithms [30], single positive label learning [26] and partial-label learning [29] by using the label shift become a common means, which is represented by the label enhancement method [21, 25]. Label enhancement (LE) employs both the feature space of the samples and the logical label space to generate a high-quality label distribution [7], which is then enforced as a regularization term on a variety of learning paradigms (see Figure 1(a)). Despite impressive achievements, current LE algorithms have yet to deliver consistent performance across various benchmarks. Regarding this issue, we introduce two insights: 1) Many LE algorithms for label distribution generation assume equal importance between the feature space and the logical label space, suggesting they make an equivalent impact. Indeed, the contribution of their features is not fair. 2) Some LE datasets include noisy and inaccurate labels, potentially resulting in significantly varied outputs for analogous inputs.

For the first insight, we give statistical evidence (see Figure 1(b)) that the difficulty of projecting the feature space to the label distribution and the logical label space to the label distribution is different on multiple benchmarks. Note that not in all datasets, logical label space moves to label distribution space at the least cost. Here, the evaluated method considers the magnitude of the energy [11] of the feature space X and the logical label space L projected to the label distribution space D with the help of the matrix W, i.e., the matrix energy of W (the sum of the product of the matrix eigenvalues and eigenvectors[1]). W can be regarded as a cost matrix, and its higher energy indicates that the projection of the matrix is more difficult, that is, the original matrix is further away from the target matrix. Fig. 1(b) illustrates that the challenge of transforming logical labels $\rightarrow$ label distributionis significantly lower than that of feature space $\rightarrow$ label distribution across the majority of datasets.

---

[1]https://wikiless.org/wiki/Eigenvalues_and_eigenvectors

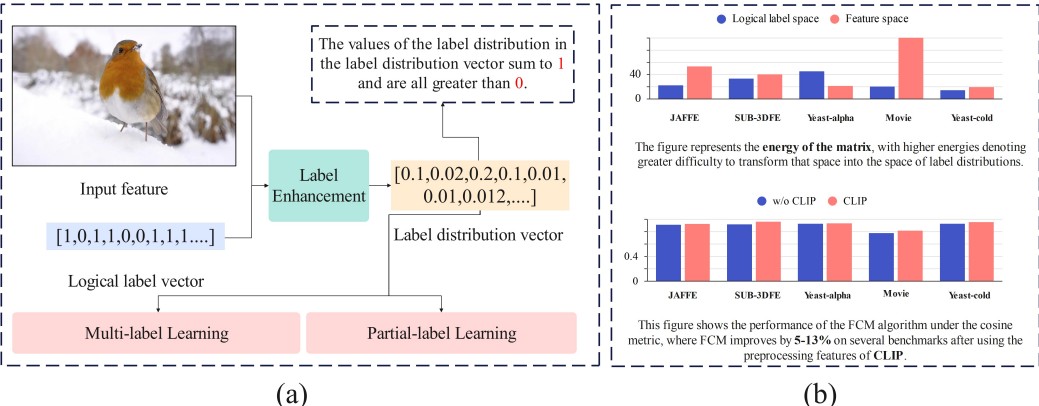

Figure 1: This figure demonstrates the role of the label enhancement paradigm and also shows our motivation for why it is necessary to re-represent information from the feature space and the logical label space.

Current LE algorithms [6, 12, 13, 15, 22, 24, 27, 32] overlook this distinction. Indeed, this oversight often results in these algorithms generating label distributions that are biased towards either the feature space or the logical label space, which hinders their ability to perform universally well across various benchmark datasets (particularly when the feature space transformation is less complex). For the second insight, there is already a large body of literature [10, 14, 33, 34] supporting that these datasets contain noise and incorrect labels. These works primarily address the issue by identifying and tolerating noisy data in the label distribution space, without considering the constraint of similar feature space inputs on the similarity of outputs in the label distribution space.

To address the challenges posed by the above two insights, we propose in this paper an LE framework based on CLIP (Contrastive Language-Image Pre-training) [20] alignment. First, we develop a pre-processing algorithm that harmonizes the transformation difficulty between the feature space and the logical label space, with the assistance of CLIP. In this context, we consider feature space vectors and logical label vectors as distinct modalities that convey identical semantics, namely, the label distribution to be derived. Significantly, this paper ensures that the dimensionality of the transformed feature vectors aligns with that of the transformed logical label vectors. Second, to mitigate the impact of noisy data, we use the similarity matrix generated during the pre-training process of CLIP for constraints i.e. by optimal transmission[2] [23] to move its distribution closer to the distribution of the label distribution.

Based on the proposed novel LE framework, our contribution includes:
**i)** We propose a pre-processing algorithm that ensures fairness in the representation of the recoded feature space and the logical labeling space to help downstream LE algorithms achieve better performance.
**ii)** We propose a novel regularization method to constrain the modeling process of CLIP to ensure that the semantic information of new features is not corrupted. Extensive experimental results and discussions demonstrate the importance of fairness representation, and there is also a significant performance improvement for downstream LE algorithms.

## 2 CLIP2LE FRAMEWORK

### 2.1 ASSUMPTIONS AND NOTATION

The concept of LE is first derived from label distribution learning [24]. The purpose of LE is to recover label distributions from logical labels as a novel type of supervised information that serves various learning paradigms. Currently, most LE algorithms can be divided into two categories, one is algorithmic adaptation [6, 7, 13] and the other is specialized algorithms [16, 22, 24]. These algorithms not only achieved SOTA results on label distribution learning, but also achieved impressive perfor-

---

[2]https://github.com/PythonOT/POT

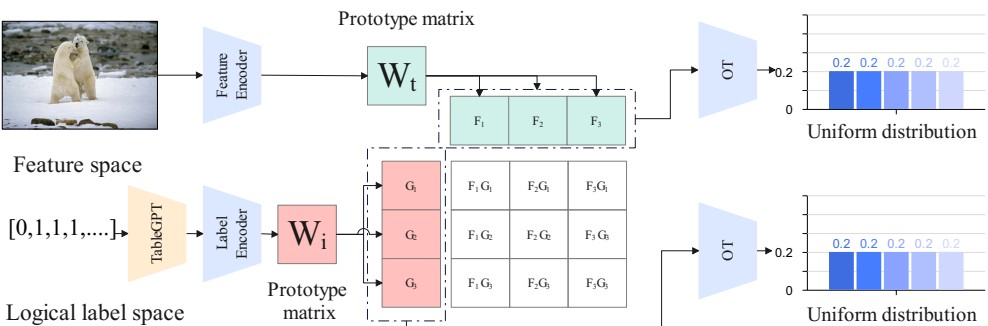

Figure 2: This figure shows the framework of our method. Our method has two main components: first, it starts with the feature space and the logical label space for contrast learning; second, the coding distributions of the feature space and the coding distributions of the logical label space are both close to the uniform distribution. The OT in the figure denotes the optimal transmission algorithm; the aim is to have relatively smooth features after encoding. $\mathbf{W}_i$ and $\mathbf{W}_t$ denote the prototype matrices which ensure that the output signals are in the same semantic space.

mance on multi-label learning [30], partial-label learning [29], and single positive-label learning [26]. Nonetheless, the existence of biased characterizations and noise within these benchmarks prevents LE algorithms from achieving consistently superior performance. We introduce a novel pre-processing framework, termed CLIP2LE, designed to supply high-quality features to subsequent LE algorithms.

**Assumptions.** We have three critical assumptions for a CLIP2LE framework (see Figure 2).

**a)** We recognize the feature space and the logical label space as inherently misaligned modalities, which can be reconciled using the assistance of CLIP. Moreover, fortunately, a large part of these benchmarks are tabular data, and the model can gobble up a large number of samples in a single batch, which ensures the performance of CLIP.

**b)** We identify a portion of the dataset that has noise and incorrect labels in the logical label space, which may lead to instability in the output of the LE model, and CLIP ensures that similar inputs generate similar outputs.

**(c)** Since both the feature space and the logical label space point to the semantics of the label distribution, their similarity should also be consistent with the distributional morphology of the label distribution. Here, we use optimal transmission to ensure the robustness of CLIP modeling.

**Notation.** Given a particular instance, the goal of our method is to learn the degree to which each label describes that instance. Input matrix $\mathcal{X} \in \mathbb{R}^{M \times N}$, where $M$ is the number of instances and $N$ is the dimension of features. $x_i$ is the $i$-th instance in the dataset. The label distribution space is defined as $\mathcal{Y} \in \mathbb{R}^{M \times L}$, where $y_j$ represents the $j$-th label. For each instance $x_i$, we define its label distribution $\mathcal{D}_i = \left\{ d_{x_i}^{y_1}, d_{x_i}^{y_2}, ..., d_{x_i}^{y_L} \right\}$, where $d_{x_i}^{y_j}$ is the description degree of the label $y_j$ for the instance $x_i$. The $d_{x_i}^{y_j}$ is constrained by $d_{x_i}^{y_j} \in [0, 1]$ and $\sum_{j=1}^{L} d_{x_i}^{y_j} = 1$. The label distribution that is predicted by the model is defined as $\mathcal{L}_i = \left\{ l_{x_i}^{y_1}, l_{x_i}^{y_2}, ..., l_{x_i}^{y_L} \right\}$. The logical label is defined as $\mathcal{G}_i = \left\{ g_{x_i}^{y_1}, g_{x_i}^{y_2}, ..., g_{x_i}^{y_L} \right\}$, where $g_{x_i}^{y_j} \in \{0, 1\}$.

## 2.2 METHOD

First, given a pairwise feature space vector $x_i$ and a logical labeling vector $\mathcal{G}_i$, we input $x_i$ and $\mathcal{G}_i$ into the feature encoder **FE** and the logical label encoder **LLE**, respectively, to obtain the re-encoded features $\mathbf{f}_i$ and $\mathbf{l}_i$. Here, both **FE** and **LLE** use MLP as an encoder where the activation function uses ReLU. Next, we conduct joint embedding and regularization for features $\mathbf{f}_i$ and $\mathbf{l}_i$. This step can be written as follows:

$$\mathbf{I}_e = \text{Normalize}(\mathbf{f}_i \mathbf{W}_i), \mathbf{T}_e = \text{Normalize}(\mathbf{l}_i \mathbf{W}_t), \tag{1}$$

where $\mathbf{W}_i$ and $\mathbf{W}_t$ denote affine matrices (prototype matrix) and Normalize denotes normalization. Finally, we compute the cosine similarity between features ($\mathbf{I}_e$, $\mathbf{T}_e$) and perform a contrast loss function. This step can be written as follows:

$$\text{logits} = \mathbf{I}_e \mathbf{T}_e^{\mathrm{T}} \times \mathrm{E}^t, \tag{2}$$

$$\text{loss} = (\text{CEL}(\text{logits}, \text{dim=0}) + \text{CEL}(\text{logits}, \text{dim=1}))/2, \quad (3)$$

where CEL denotes the cross-entropy function, $t$ denotes temperature parameters for index E, and dim=0 and dim=1 denote computations done in a given dimension, respectively.

**Feature distribution homogenization.** So far, we expect features $\mathbf{f}_i$ and $\mathbf{l}_i$ to be smooth, which facilitates fast convergence of the downstream LE model during training. In general, we can use distribution measure formulas such as KL dispersion, J dispersion, etc., however, they are computationally expensive. Here, we use optimal transmission (OT) to make the distributions of $\mathbf{f}_i$ and $\mathbf{l}_i$ close to the uniform distribution at a small cost. Specifically, first, we need to compute the migration cost of these two distributions; in this paper, we use earth mover's distance (EMD)[3]. Next, we use Sinkhorn [5] to optimize this transmission cost. This step can be written as follows:

$$\text{Sinkhorn} \to \text{EMD}(\mathbf{f}_i, \mathcal{U}), \text{Sinkhorn} \to \text{EMD}(\mathbf{l}_i, \mathcal{U}), \quad (4)$$

where Sinkhorn denotes the optimization method, $\mathcal{U}$ denotes uniform distribution, and here the hyperparameter we set to the $10^{-3}$.

**High-quality feature generation.** After N iterations we obtain high quality $\mathbf{f}_i$ and $\mathbf{l}_i$. For the range of values of $\mathbf{f}_i$ and $\mathbf{l}_i$, we use Tanh for constraints to avoid outliers. These two new features are used in downstream LE algorithms for inference to obtain accurate label distributions $\mathcal{L}_i$.

**Logical label vectors with the help of TableGPT.** Predictably, we struggle with treating the logical label vector as a modality, due to the fact that the logical label vector contains less information and is sparser compared to the feature space $\mathcal{X}$. Simply increasing the neurons and depth of the **LLE** can lead to pre-training that is difficult to converge and an imbalance between **LLE** and **FE** exists. To address this problem, we attempt to introduce a prior on the large model (TableGPT) to precode the logistic label vectors. We input it as tabular data into TableGPT's Table Encode to obtain a high-quality representation. It is worth noting that we are frozen the parameters of TableGPT during the training process. We also try to input the logic label vectors into BERT as text, but it doesn't work well. Here, we extracted the 256-dimensional vectors of the middle layer in Table Encoder as new feature inputs to the MLP with only 3 layers.

**Construction of the prototype matrix $\mathbf{W}_i$ and $\mathbf{W}_t$.** To make the output space of **FE** and **LLE** more compact, we construct a prototype matrix as the projection space. The prototype space consists of a set of representative vectors. Specifically, we begin by splicing the input space $\mathcal{X}$ and the logical label space $\mathbf{G}$ into a matrix $\mathbf{Y} \in \mathbb{R}^{M \times (N+L)}$. Next, we use the nearest neighbor algorithm to obtain $L$ clusters, the center of which is the representation vector. Finally, we concatenate these $L$ cluster centers into a matrix $\hat{\mathbf{W}}$ and use PCA to compress it to the required dimensions. The whole step can be written as:

$$\mathbf{W}_i = \text{PCA}(\text{CONCAT}(\text{KNN}(\text{SPL}(\mathcal{X}, \mathbf{G})))), \quad (5)$$

where $\mathbf{W}_i$ and $\mathbf{W}_t$ are constructed in the same way.

## 3 EXPERIMENTS

Extensive experiments are conducted in this section to verify the effectiveness and competitiveness of CLIP2LE. We performed four sets of experiments demonstrating the impact of our approach on downstream LE algorithms, including the impact of our algorithms enforced on LE algorithms on label distribution learning, multi-label learning, partial-label learning, and single positive multi-label learning. In addition, we performed experiments evaluating CLIP2LE's robustness against noise and the effect of other CLIP algorithms (MaskCLIP and MetaCLIP) on the experiments.

### 3.1 LE ALGORITHMS ON LABEL DISTRIBUTION LEARNING

**Experimental setup.** We use 13 real-world datasets [18, 24] for evaluation[4]. Our algorithm serves as a pre-processing algorithm for the 8 LE methods, including FCM [6], KM [13], LP [15], ML [12], GLLE [24], LESC [22], LIB [32], and LEVI [27]. To evaluate the recovery performance, we adopt 6 metrics, namely Chebyshev, Canberra, Clark, Kullback-Leibler, Cosine, and Intersection [7, 22, 24].

---

[3]https://encyclopedia.thefreedictionary.com/Earth+Mover%27s+Distance

[4]http://palm.seu.edu.cn/xgeng/LDL/index.htm

Here, we show only the performance of the metric functions Chebyshev and Clark. The rest is shown in the Supplementary Material.

| Metric | Chebyshev ↓ | | | | | | | | Clark ↓ | | | | | | | |
|---|---|---|---|---|---|---|---|---|---|---|---|---|---|---|---|---|
| Method | FCM | KM | LP | ML | GLLE | LESC | LEVI | LIB | FCM | KM | LP | ML | GLLE | LESC | LEVI | LIB |
| Movie | 0.230 | 0.234 | 0.161 | 0.164 | 0.122 | 0.121 | 0.110 | **0.107** | 0.859 | 1.766 | 0.913 | 1.140 | 0.569 | 0.564 | 0.551 | **0.517** |
| SUB-3DFE | 0.135 | 0.238 | 0.123 | 0.233 | 0.126 | 0.122 | 0.095 | **0.094** | 0.482 | 1.907 | 0.580 | 1.848 | 0.391 | 0.378 | 0.303 | **0.297** |
| JAFFE | 0.132 | 0.214 | 0.107 | 0.186 | 0.087 | **0.069** | 0.075 | 0.071 | 0.522 | 1.874 | 0.502 | 1.519 | 0.377 | 0.276 | 0.290 | **0.262** |
| Yeast-alpha | 0.044 | 0.063 | 0.040 | 0.057 | 0.020 | 0.015 | **0.012** | 0.017 | 0.821 | 3.153 | 1.185 | 3.088 | 0.337 | **0.253** | 0.319 | 0.275 |
| Yeast-cdc | 0.051 | 0.076 | 0.042 | 0.071 | 0.022 | 0.019 | **0.016** | 0.017 | 0.739 | 2.885 | 1.014 | 2.825 | 0.306 | 0.251 | 0.323 | **0.242** |
| Yeast-cold | 0.141 | 0.252 | 0.137 | 0.242 | 0.066 | 0.056 | 0.082 | **0.054** | 0.433 | 1.472 | 0.503 | 1.440 | 0.176 | 0.152 | 0.269 | **0.146** |
| Yeast-diau | 0.124 | 0.152 | 0.099 | 0.148 | 0.053 | **0.042** | 0.044 | 0.049 | 0.838 | 1.886 | 0.788 | 1.844 | 0.296 | **0.224** | 0.295 | 0.273 |
| Yeast-dtt | 0.097 | 0.257 | 0.128 | 0.244 | 0.052 | 0.043 | 0.084 | **0.034** | 0.329 | 1.477 | 0.499 | 1.446 | 0.143 | 0.119 | 0.294 | **0.092** |
| Yeast-elu | 0.052 | 0.078 | 0.044 | 0.072 | 0.023 | 0.019 | **0.017** | 0.018 | 0.579 | 2.768 | 0.973 | 2.711 | 0.295 | 0.241 | 0.317 | **0.224** |
| Yeast-heat | 0.169 | 0.175 | 0.086 | 0.165 | 0.049 | 0.046 | 0.052 | **0.039** | 0.580 | 1.802 | 0.568 | 1.764 | 0.213 | 0.199 | 0.288 | **0.165** |
| Yeast-spo | 0.130 | 0.175 | 0.090 | 0.171 | 0.062 | 0.060 | 0.055 | **0.053** | 0.520 | 1.811 | 0.558 | 1.768 | 0.266 | 0.258 | 0.277 | **0.224** |
| Yeast-spo5 | 0.162 | 0.277 | 0.114 | 0.273 | 0.099 | 0.092 | 0.091 | **0.076** | 0.395 | 1.059 | 0.274 | 1.036 | 0.197 | 0.185 | 0.209 | **0.158** |
| Yeast-sopem | 0.233 | 0.408 | 0.163 | 0.403 | 0.088 | 0.087 | 0.115 | **0.069** | 0.401 | 1.028 | 0.272 | 1.004 | 0.132 | 0.129 | 0.182 | **0.104** |

| Metric | Chebyshev ↓ | | | | | | | | Clark ↓ | | | | | | | |
|---|---|---|---|---|---|---|---|---|---|---|---|---|---|---|---|---|
| Ours + Method | FCM | KM | LP | ML | GLLE | LESC | LEVI | LIB | FCM | KM | LP | ML | GLLE | LESC | LEVI | LIB |
| Movie | 0.202 | 0.214 | 0.150 | 0.162 | 0.112 | 0.102 | 0.100 | **0.095** | 0.723 | 1.500 | 0.843 | 0.987 | 0.500 | 0.533 | 0.625 | **0.401** |
| SUB-3DFE | 0.121 | 0.209 | 0.104 | 0.222 | 0.131 | 0.100 | 0.099 | **0.082** | 0.481 | 1.807 | 0.410 | 1.222 | 0.335 | 0.312 | 0.250 |  |
| JAFFE | 0.112 | 0.198 | 0.095 | 0.155 | 0.090 | 0.066 | **0.056** | 0.069 | 0.521 | 1.588 | 0.487 | 1.322 | **0.229** | 0.256 | 0.232 | **0.229** |
| Yeast-alpha | 0.032 | 0.059 | 0.033 | 0.050 | 0.019 | 0.012 | **0.010** | **0.010** | 0.810 | 2.998 | 1.010 | 3.095 | 0.320 | **0.223** | 0.304 | 0.224 |
| Yeast-cdc | 0.050 | 0.056 | 0.041 | 0.080 | 0.010 | 0.018 | 0.015 | **0.010** | 0.539 | 2.288 | 0.982 | 2.547 | 0.237 | **0.222** | 0.310 | 0.238 |
| Yeast-cold | 0.123 | 0.233 | 0.106 | 0.214 | 0.055 | 0.051 | 0.088 | **0.050** | 0.399 | 1.256 | 0.666 | 1.425 | 0.168 | 0.110 | 0.260 | **0.139** |
| Yeast-diau | 0.111 | 0.143 | 0.095 | 0.144 | 0.052 | **0.041** | 0.049 | 0.050 | 0.776 | 1.566 | 0.789 | 1.654 | **0.122** | 0.205 | 0.300 | 0.266 |
| Yeast-dtt | 0.085 | 0.220 | 0.113 | 0.236 | 0.046 | 0.042 | 0.069 | **0.031** | 0.320 | 1.289 | 0.490 | 1.255 | 0.145 | 0.118 | 0.302 | **0.090** |
| Yeast-elu | 0.044 | 0.065 | 0.041 | 0.053 | 0.020 | **0.013** | 0.014 | 0.015 | 0.454 | 2.133 | 0.965 | 2.551 | 0.243 | 0.240 | 0.328 | **0.225** |
| Yeast-heat | 0.132 | 0.105 | 0.056 | 0.144 | 0.041 | 0.045 | 0.050 | **0.035** | 0.582 | 1.602 | 0.556 | 1.789 | 0.224 | 0.190 | 0.206 | **0.160** |
| Yeast-spo | 0.129 | 0.165 | 0.088 | 0.165 | 0.058 | 0.061 | 0.056 | **0.050** | 0.521 | 1.800 | 0.541 | 1.413 | 0.260 | 0.260 | 0.256 | **0.220** |
| Yeast-spo5 | 0.152 | 0.272 | 0.110 | 0.256 | 0.090 | 0.091 | 0.095 | **0.081** | 0.390 | 1.001 | 0.257 | 0.999 | 0.195 | 0.180 | 0.211 | **0.108** |
| Yeast-sopem | 0.253 | 0.308 | 0.156 | 0.456 | 0.066 | 0.088 | 0.103 | **0.059** | 0.421 | 0.967 | 0.256 | 1.000 | 0.135 | 0.102 | 0.156 | **0.100** |

Table 1: Recovery results on 13 real-world datasets. ↓ indicates that "the smaller the better" and ↑ means that "the larger the better". The following two table blocks represent the 8 LE algorithms using our pre-processing steps. We highlight the best recovery results.

**Result.** We provide the detailed comparison results on 13 real-world datasets in Table 1. Overall, we found two interesting results: 1) our approach does not allow all algorithms to improve their performance, and 2) our approach is very friendly to non-deep algorithms, especially adaption ones. In addition, for the LIB algorithm, the ability of our method to help it boost is effective, probably due to the fact that the model capacity of LIB reaches its upper limit.

## 3.2 LE ALGORITHMS ON MULTI-LABEL LEARNING

**Experimental setup.** In this subsection, the efficiency and the performance of FLEM [31] + our method are evaluated in multiple multi-label learning datasets. All methods are implemented by PyTorch. All the computations are performed on a GPU server with NVIDIA Tesla V100. We use eight datasets, including two text datasets and six image datasets.

| Metric | Hamming loss ↓ | | | | | | | | Ranking loss↓ | | | | | | | |
|---|---|---|---|---|---|---|---|---|---|---|---|---|---|---|---|---|
| Dataset | AAPD | Reuters | VOC07 | VOC12 | COCO14 | COCO17 | CUB | NUS | AAPD | Reuters | VOC07 | VOC12 | COCO14 | COCO17 | CUB | NUS |
| FLEM-S | 0.0276 | 0.0040 | 0.0239 | 0.0226 | 0.0191 | 0.0207 | 0.0850 | 0.0138 | 0.0522 | 0.0092 | 0.0194 | 0.0140 | 0.0239 | 0.0286 | 0.1058 | 0.0149 |
| FLEM-T | 0.0271 | 0.0040 | 0.0243 | 0.0228 | 0.0192 | 0.0209 | 0.0849 | 0.0139 | 0.0483 | 0.0094 | 0.0199 | 0.0138 | 0.0238 | 0.0286 | 0.1048 | 0.0148 |
| FLEM-D | 0.0271 | 0.0041 | 0.0237 | 0.0231 | 0.0192 | 0.0209 | 0.0848 | 0.0137 | 0.0432 | 0.0091 | 0.0189 | 0.0139 | 0.0231 | 0.0279 | 0.1039 | 0.0139 |

| Metric | Hamming loss ↓ | | | | | | | | Ranking loss ↓ | | | | | | | |
|---|---|---|---|---|---|---|---|---|---|---|---|---|---|---|---|---|
| Dataset | AAPD | Reuters | VOC07 | VOC12 | COCO14 | COCO17 | CUB | NUS | AAPD | Reuters | VOC07 | VOC12 | COCO14 | COCO17 | CUB | NUS |
| FLEM-S + our method | 0.0265 | 0.0038 | 0.0221 | 0.0220 | 0.0190 | 0.0203 | 0.0843 | 0.0139 | 0.0521 | 0.0098 | 0.0156 | 0.0131 | 0.0233 | 0.0285 | 0.1050 | 0.0145 |
| FLEM-T + our method | 0.0263 | 0.0043 | 0.0240 | 0.0222 | 0.0190 | 0.0213 | 0.0840 | 0.0135 | 0.0480 | 0.0093 | 0.0193 | 0.0122 | 0.0245 | 0.0283 | 0.1021 | 0.0140 |
| FLEM-D + our method | 0.0230 | 0.0033 | 0.0230 | 0.0221 | 0.0196 | 0.0201 | 0.0801 | 0.0133 | 0.0430 | 0.0090 | 0.0185 | 0.0130 | 0.0234 | 0.0278 | 0.1033 | 0.0121 |

Table 2: Recovery results on 8 real-world datasets. ↓ indicates that "the smaller the better". The following two table blocks represent the 8 LE algorithms using our preprocessing steps.

**Result.** We provide the detailed comparison results on 8 real-world datasets in Table 2. Overall, we observe that our method does not uniformly enhance the performance of FLEM, particularly in scenarios with a fewer number of labels. Conversely, our approach demonstrates a substantial improvement in cases where there is a higher number of labels.

### 3.3 LE ALGORITHMS ON PARTIAL LABEL LEARNING

Our method imposes an evaluation on PL-LE [25]. First, our method + PL-LE and PL-LE were evaluated on 168 UCI data sets [1]. Our method + PL-LE achieves comparable performance against PL-LE in 92.0% cases (150 out of 168) and 90.1% cases (147 out of 168) respectively. In addition, PL-LE achieves superior performance against PL-LEAF and PL-ECOC in 8.0% cases (18 out of 168) and 9.9% cases (21 out of 168) respectively.

Second, our method + PL-LE and PL-LE were evaluated in Real-World Data Sets. On all data sets, our method + PL-LE achieves superior or at least comparable performance against PL-LE. Our method improves 2%, 3.2%, 4.1%, 5.9%, 2.9%, and 3.3% compared to PL-LE on the FG-NET [19], Lost [4], MSRCv2 [17], BirdSong [3], Soccer Player [28], and Yahoo! News [8] dataset, respectively.

### 3.4 LE ALGORITHMS ON SINGLE-POSITIVE MULTI-LABEL LEARNING

**Experimental setup.** In the experiments, we adopt twelve widely-used multi-label learning datasets [9], which cover a broad range of cases with diversified multi-label properties. To evaluate the performance of our method + SPMLL [26] methods, we generate the single positive training data by randomly selecting one positive label to keep for each training example in the multi-label learning datasets. For each dataset, we run the comparing methods with 80%/10%/10% train/validation/test split. The validation and test sets are always fully labeled. Five popular multi-label metrics *Ranking loss*, *Hamming loss*, *One-error*, *Coverage*, and *Average precision* are employed for performance evaluation. Furthermore, for *Average precision*, the *larger* the values the better the performance. While for the other four metrics, the *smaller* the values the better the performance.

**Result.** Our method compared to SPMLL on CAL500, image, scene, yeast, core15k, revl-s1, core116k-s1, delicious, iaprtc12, espgame, mirfiickr, tmc2007 dataset improves 3%, 4.1%, 1.0%, 0.92%, 8.9%, 12.6%, 1.1%, -2.1%, 3.9%, 4.4%, 15.8%, 6.6%, respectively, where the negative sign denotes performance degradation.

| Metric | Chebyshev ↓ | | | | | | | | Clark ↓ | | | | | | | |
|---|---|---|---|---|---|---|---|---|---|---|---|---|---|---|---|---|
| w/o AS(i) + Method | FCM | KM | LP | ML | GLLE | LESC | LEVI | LIB | FCM | KM | LP | ML | GLLE | LESC | LEVI | LIB |
| Movie | 0.211 | 0.223 | 0.162 | 0.165 | 0.110 | 0.129 | 0.118 | **0.098** | 0.729 | 1.654 | 0.855 | 0.990 | 0.523 | 0.546 | 0.657 | **0.422** |
| SUB-3DFE | 0.120 | 0.210 | 0.105 | 0.228 | 0.141 | 0.133 | 0.102 | **0.091** | 0.499 | 1.944 | 0.430 | 1.229 | **0.221** | 0.339 | 0.310 | 0.290 |
| JAFFE | 0.113 | 0.201 | 0.123 | 0.166 | 0.091 | 0.063 | **0.059** | 0.077 | 0.543 | 1.589 | 0.498 | 1.400 | 0.222 | 0.299 | 0.243 | 0.230 |
| Yeast-alpha | 0.031 | 0.069 | 0.048 | 0.052 | 0.033 | 0.033 | **0.011** | 0.011 | 0.859 | 2.997 | 1.163 | 3.421 | 0.321 | **0.220** | 0.395 | 0.266 |
| Yeast-cdc | 0.059 | 0.068 | 0.055 | 0.093 | 0.012 | 0.025 | 0.033 | **0.012** | 0.599 | 2.489 | 1.177 | 2.643 | **0.221** | 0.255 | 0.319 | 0.256 |
| Yeast-cold | 0.124 | 0.289 | 0.133 | 0.225 | 0.059 | 0.056 | 0.087 | **0.052** | 0.455 | 1.432 | 0.678 | 1.566 | 0.232 | 0.115 | 0.261 | **0.144** |

Table 3: We removed the effect of homogenization of the feature distribution. Recovery results on 6 real-world datasets. ↓ indicates that "the smaller the better". We highlight the best recovery results.

| Metric | Chebyshev ↓ | | | | | | | | Clark ↓ | | | | | | | |
|---|---|---|---|---|---|---|---|---|---|---|---|---|---|---|---|---|
| w/o AS(ii) + Method | FCM | KM | LP | ML | GLLE | LESC | LEVI | LIB | FCM | KM | LP | ML | GLLE | LESC | LEVI | LIB |
| Movie | 0.241 | 0.252 | 0.174 | 0.165 | 0.129 | 0.120 | **0.109** | 0.110 | 0.887 | 1.767 | 0.947 | 1.156 | 0.570 | 0.561 | 0.587 | **0.512** |
| SUB-3DFE | 0.136 | 0.289 | 0.197 | 0.265 | 0.127 | 0.125 | 0.096 | **0.093** | 0.483 | 1.905 | 0.581 | 1.899 | 0.399 | 0.379 | 0.302 | **0.299** |
| JAFFE | 0.133 | 0.212 | 0.175 | 0.187 | 0.089 | **0.071** | 0.079 | 0.078 | 0.521 | 1.844 | 0.519 | 1.566 | 0.397 | 0.296 | 0.292 | **0.260** |
| Yeast-alpha | 0.049 | 0.065 | 0.041 | 0.055 | 0.029 | 0.033 | **0.027** | 0.065 | 0.820 | 3.159 | 1.192 | 3.222 | 0.322 | **0.250** | 0.336 | 0.278 |
| Yeast-cdc | 0.055 | 0.077 | 0.047 | 0.067 | 0.023 | 0.019 | 0.021 | **0.018** | 0.741 | 2.889 | 1.055 | 2.899 | 0.316 | 0.255 | 0.325 | **0.241** |
| Yeast-cold | 0.145 | 0.300 | 0.139 | 0.241 | 0.069 | 0.058 | 0.089 | **0.056** | 0.435 | 1.477 | 0.615 | 1.444 | 0.179 | 0.172 | 0.272 | **0.145** |

Table 4: We removed the effects of TabGP. Recovery results on 6 real-world datasets. ↓ indicates that "the smaller the better". We highlight the best recovery results.

| Metric | Chebyshev ↓ | | | | | | | | Clark ↓ | | | | | | | |
|---|---|---|---|---|---|---|---|---|---|---|---|---|---|---|---|---|
| w/o AS(iii) + Method | FCM | KM | LP | ML | GLLE | LESC | LEVI | LIB | FCM | KM | LP | ML | GLLE | LESC | LEVI | LIB |
| Movie | 0.249 | 0.267 | 0.173 | 0.195 | 0.177 | 0.121 | **0.111** | 0.119 | 0.889 | 1.766 | 0.933 | 1.157 | 0.571 | 0.569 | 0.582 | **0.519** |
| SUB-3DFE | 0.137 | 0.292 | 0.195 | 0.264 | 0.123 | 0.133 | 0.097 | **0.095** | 0.485 | 1.910 | 0.589 | 1.912 | 0.397 | 0.388 | 0.313 | **0.298** |
| JAFFE | 0.135 | 0.213 | 0.177 | 0.188 | 0.091 | **0.070** | 0.075 | 0.079 | 0.523 | 1.842 | 0.588 | 1.567 | 0.393 | 0.294 | 0.293 | **0.261** |
| Yeast-alpha | 0.056 | 0.066 | 0.043 | 0.057 | 0.032 | 0.035 | **0.022** | 0.069 | 0.829 | 3.177 | 1.193 | 3.229 | 0.325 | **0.251** | 0.339 | 0.289 |
| Yeast-cdc | 0.056 | 0.078 | 0.055 | 0.069 | 0.021 | 0.015 | 0.029 | **0.011** | 0.745 | 2.894 | 1.053 | 2.895 | 0.317 | 0.259 | 0.321 | **0.242** |
| Yeast-cold | 0.147 | 0.314 | 0.135 | 0.245 | 0.072 | 0.063 | 0.094 | **0.059** | 0.436 | 1.466 | 0.619 | 1.474 | 0.184 | 0.173 | 0.274 | **0.144** |

Table 5: We remove the effect of the prototype matrix. Recovery results on 6 real-world datasets. ↓ indicates that "the smaller the better". We highlight the best recovery results.

## 3.5 ABLATION STUDY FOR CLIP2LE

To evaluate the effectiveness of our method, we conduct 3 experiments on the 3.1 task.

**i) Remove the feature distribution homogenization.** We remove the loss term that moves closer to the uniform distribution. As shown in Table 3, we find that the performance degradation is significant when the number of labels is greater than 6. This may be due to the principle of label distribution learning that as the number of labels increases, the lower the percentage of each label, the closer it is to a uniform distribution.

**ii) Remove the TableGPT.** To evaluate the effectiveness of recoding the logical label space by TableGPT, we removed TableEncoder. As shown in Table 4, we find that the performance degradation is significant when the number of labels is smaller than 6.

**iii) Remove the prototype matrix.** To demonstrate the effectiveness of the prototype matrices, the prototype matrices in our approach are replaced by learnable parameter matrices. As shown in Table 5, we find that the performance of the prototype matrix is almost homotopic to that of TableGPT and that the prototype matrix may be more important for labeling the role of logical vector encoding.

| Metric | | | | Chebyshev ↓ | | | | | | | | Clark ↓ | | | |
|---|---|---|---|---|---|---|---|---|---|---|---|---|---|---|---|
| Method without our method | FCM | KM | LP | ML | GLLE | LESC | LEVI | LIB | FCM | KM | LP | ML | GLLE | LESC | LEVI | LIB |
| Movie | 0.231 | 0.236 | 0.169 | 0.177 | 0.125 | 0.126 | 0.112 | **0.107** | 0.869 | 1.777 | 0.921 | 1.144 | 0.570 | 0.576 | 0.555 | **0.519** |
| SUB-3DFE | 0.136 | 0.244 | 0.126 | 0.236 | 0.123 | 0.127 | 0.100 | **0.096** | 0.491 | 1.913 | 0.589 | 1.896 | 0.393 | 0.379 | 0.311 | **0.298** |
| Metric | | | | Chebyshev ↓ | | | | | | | | Clark ↓ | | | |
| Method with our method | FCM | KM | LP | ML | GLLE | LESC | LEVI | LIB | FCM | KM | LP | ML | GLLE | LESC | LEVI | LIB |
| Movie | 0.239 | 0.248 | 0.178 | 0.199 | 0.165 | 0.133 | 0.119 | **0.108** | 0.887 | 1.995 | 0.932 | 1.142 | 0.599 | 0.601 | 0.555 | **0.544** |
| SUB-3DFE | 0.164 | 0.278 | 0.155 | 0.280 | 0.156 | 0.133 | 0.125 | **0.090** | 0.495 | 1.966 | 0.589 | 1.879 | 0.466 | 0.372 | 0.333 | **0.299** |

Table 6: Recovery results on 2 real-world datasets. ↓ indicates that "the smaller the better". We highlight the best recovery results.

| Metric | | | | Chebyshev ↓ | | | | | | | | Clark ↓ | | | |
|---|---|---|---|---|---|---|---|---|---|---|---|---|---|---|---|
| Method without our method | FCM | KM | LP | ML | GLLE | LESC | LEVI | LIB | FCM | KM | LP | ML | GLLE | LESC | LEVI | LIB |
| Movie | 0.298 | 0.246 | 0.255 | 0.198 | 0.213 | 0.189 | 0.144 | **0.143** | 0.956 | 1.989 | 0.977 | 1.321 | 0.689 | 0.762 | 0.699 | **0.688** |
| SUB-3DFE | 0.157 | 0.399 | 0.245 | 0.319 | 0.256 | 0.277 | 0.189 | **0.149** | 0.569 | 2.005 | 0.789 | 2.449 | 0.457 | 0.444 | 0.372 | **0.333** |
| Metric | | | | Chebyshev ↓ | | | | | | | | Clark ↓ | | | |
| Method without our method | FCM | KM | LP | ML | GLLE | LESC | LEVI | LIB | FCM | KM | LP | ML | GLLE | LESC | LEVI | LIB |
| Movie | 0.275 | 0.230 | 0.242 | 0.143 | 0.202 | 0.165 | 0.140 | **0.122** | 0.901 | 1.566 | 0.879 | 1.132 | 0.544 | 0.712 | 0.633 | **0.540** |
| SUB-3DFE | 0.152 | 0.347 | 0.201 | 0.311 | 0.206 | 0.223 | 0.175 | **0.137** | 0.555 | 1.998 | 0.753 | 2.407 | 0.438 | 0.408 | 0.370 | **0.300** |

Table 7: Recovery results on 2 real-world datasets. ↓ indicates that "the smaller the better". We highlight the best recovery results. These two datasets were injected with $0.5\times$ Gaussian noise.

## 3.6 ROBUSTNESS EVALUATION OF CLIP2LE

To evaluate the ability of our method to tolerate noise, we present two experiments: one in which the labels in the logical label space are incorrect; and the other in which the logical label space carries Gaussian noise.

1) Since we assumed the problem of incorrect labels in the space of label distributions $\mathcal{D}$, we imposed 20% incorrect labels (for example, [0.1, 0.2. 0.7] → [0.7, 0.1, 0.2]) on the training set in task 3.1 to validate the effectiveness of our method. As shown in Table 6, the interference of incorrect labels can be effectively eliminated by our method compared to the method without using preprocessing.

2) As shown in Table 7, we inject 50% Gaussian noise ($0.5\times$) into the label labeling space, and the LE algorithm using CLIP2LE has better performance.

## 3.7 PERFORMANCE OF DOWNSTREAM TASKS

To evaluate the effect of using the CLIP2LE method on the downstream prediction task, we select the LIB and LEVI algorithms as a baseline to be assessed on the SJAFFE dataset. Since the SJAFFE dataset is a face classification task, we employ ResNet-18 for prediction; where the cross-entropy function is used as the main loss term, and the label distributions predicted by LIB and LEVI are used as the regularization terms, where the weights of the regularization terms are all set to 0.15.

We found that the accuracy of pure LIB-based and LEVI-assisted prediction of facial emotion is 95.2% and 94.9%, respectively. While the accuracy of LIB and LEVI-assisted ResNet-18 prediction with CLIP2LE is 96.6% and 96.1%, respectively. It is worth noting that the prediction accuracy of ResNet-18 for facial emotions is 93.9%, without considering the regularization term. Here, our epochs are set to 200, the learning rate is 0.005, the batch size is set to 16, and the optimizer uses AdamW.

| Metric | | Chebyshev ↓ | | | | | | | | Clark ↓ | | | | | | |
|---|---|---|---|---|---|---|---|---|---|---|---|---|---|---|---|---|
| Method without our method | FCM | KM | LP | ML | GLLE | LESC | LEVI | LIB | FCM | KM | LP | ML | GLLE | LESC | LEVI | LIB |
| Movie | 0.298 | 0.246 | 0.255 | 0.198 | 0.213 | 0.189 | 0.144 | **0.143** | 0.956 | 1.989 | 0.977 | 1.321 | 0.689 | 0.762 | 0.699 | **0.688** |
| SUB-3DFE | 0.157 | 0.399 | 0.245 | 0.319 | 0.256 | 0.277 | 0.189 | **0.149** | 0.569 | 2.005 | 0.789 | 2.449 | 0.457 | 0.444 | 0.372 | **0.333** |
| Metric | | Chebyshev ↓ | | | | | | | | Clark ↓ | | | | | | |
| Method with our method | FCM | KM | LP | ML | GLLE | LESC | LEVI | LIB | FCM | KM | LP | ML | GLLE | LESC | LEVI | LIB |
| Movie | 0.275 | 0.230 | 0.242 | 0.143 | 0.202 | 0.165 | 0.140 | **0.122** | 0.901 | 1.566 | 0.879 | 1.132 | 0.544 | 0.712 | 0.633 | **0.540** |
| SUB-3DFE | 0.152 | 0.347 | 0.201 | 0.311 | 0.206 | 0.223 | 0.175 | **0.137** | 0.555 | 1.998 | 0.753 | 2.407 | 0.438 | 0.408 | 0.370 | **0.300** |
| Method with MaskCLIP | FCM | KM | LP | ML | GLLE | LESC | LEVI | LIB | FCM | KM | LP | ML | GLLE | LESC | LEVI | LIB |
| Movie | 0.272 | 0.228 | 0.245 | 0.144 | 0.200 | 0.162 | 0.138 | **0.121** | 0.899 | 1.562 | 0.877 | 1.130 | 0.542 | 0.711 | 0.629 | **0.537** |
| SUB-3DFE | 0.151 | 0.338 | 0.202 | 0.313 | 0.201 | 0.215 | 0.170 | **0.133** | 0.554 | 1.995 | 0.742 | 2.402 | 0.433 | 0.402 | 0.365 | **0.293** |
| Method with MetaCLIP | FCM | KM | LP | ML | GLLE | LESC | LEVI | LIB | FCM | KM | LP | ML | GLLE | LESC | LEVI | LIB |
| Movie | 0.272 | 0.228 | 0.243 | 0.144 | 0.210 | 0.169 | 0.132 | **0.121** | 0.921 | 1.542 | 0.870 | 1.121 | 0.539 | 0.710 | 0.635 | **0.532** |
| SUB-3DFE | 0.149 | 0.344 | 0.199 | 0.310 | 0.205 | 0.219 | 0.170 | **0.138** | 0.523 | 1.989 | 0.740 | 2.405 | 0.433 | 0.412 | 0.369 | **0.298** |

Table 8: Recovery results on 2 real-world datasets. ↓ indicates that "the smaller the better". We highlight the best recovery results. We evaluated the effect of different CLIP versions.

## 3.8 EFFECT OF OTHER CLIP VERSIONS

To align the semantics of the logical label space and the feature space, we conduct experiments based on pure CLIP. Here, we investigate the effect of MaskCLIP and MetaCLIP on the experimental results. As shown in Table 8, MaskCLIP and MetaCLIP are essentially improved over pure CLIP on both datasets, with a decrease in very few algorithms. However, MaskCLIP and MetaCLIP consume much more computational resources than pure CLIP.

## 3.9 ESTIMATION ERROR BOUND

In this subsection, we estimate the error bounds of our method using the example of single positive multi-label learning. The empirical risk estimator (loss term for [26]) according to 3.4 can be rewritten as:

$$\mathcal{R}_{spmll}(f) = \frac{1}{n} \sum_{i=1}^{n} \sum_{j=1}^{L} \left( w_i^j \ell_i^j + \bar{w}_i^j \bar{\ell}_i^j \right),$$
(6)

where $w_i^j = \frac{d_i^j}{p(y^\gamma=1|x_i)c}$ and $\bar{w}_i^j = \frac{1-d_i^j}{p(y^\gamma=1|x_i)c}$. Then the total loss function $\mathcal{L}_{sp}$ is

$$\mathcal{L}_{sp} = \sum_{j=1}^{L} \left( w_i^j \ell_i^j + \bar{w}_i^j \bar{\ell}_i^j \right) + (\text{CEL}(\text{logits}, \text{dim=0}) + \text{CEL}(\text{logits}, \text{dim=1})) /2.$$
(7)

We define a function space as:

$$\mathcal{K}_{sp} = \left\{ (x,y) \mapsto \sum_{j=1}^{L} \left( w^j \ell^j + \bar{w}^j \bar{\ell}^j \right) + (\text{CEL}(\text{logits}, \text{dim=0}) + \text{CEL}(\text{logits}, \text{dim=1})) /2 | f \in \mathcal{F} \right\},$$
(8)

and denote the expected Rademacher complexity [2] of $\mathcal{K}_{sp}$ as:

$$\widetilde{\mathfrak{R}}_n(\mathcal{K}_{sp}) = \mathbb{E}_{x,y,\boldsymbol{\sigma}} \left[ \sup_{g \in \mathcal{G}_{sp}} \frac{1}{n} \sum_{i=1}^{n} \sigma_i g(x_i, y_i) \right],$$
(9)

where $\boldsymbol{\sigma} = \{\sigma_1, \sigma_2, \ldots, \sigma_n\}$ is $n$ Rademacher variables with $\sigma_i$ independently uniform variable taking value in $\{+1, -1\}$. We suppose that the SPMLL loss function $\mathcal{L}_{sp}$ could be bounded by $M$,

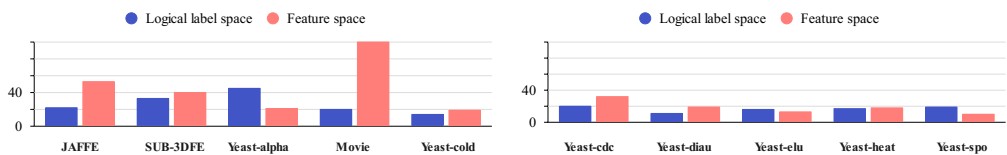

The figure represents the **energy of the matrix**, with higher energies denoting greater difficulty to transform that space into the space of label distributions.

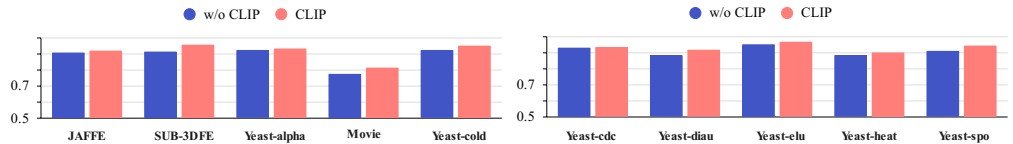

This figure shows the performance of the FCM algorithm under the cosine metric, where FCM improves by **5-13%** on several benchmarks after using the preprocessing features of **CLIP**.

Figure 3: This figure provides a comprehensive view of how difficult it is to transform information from a matrix across multiple benchmarks, with higher matrix energies representing more difficult information transformations.

i.e., $M = \sup_{x \in \mathcal{X}, f \in \mathcal{F}, y \in \mathcal{Y}} \mathcal{L}_{sp}(f(x), y)$, and for any $\delta > 0$, with probability at least $1 - \delta$, then we have

$$\sup_{f \in \mathcal{F}} R_{sp}(f) - \widehat{R}_{sp}(f) \leq 2\widetilde{\mathfrak{R}}_n(\mathcal{K}_{sp}) + \frac{M}{2}\sqrt{\frac{\log \frac{2}{\delta}}{2n}}.$$

We suppose that the loss function $\ell(f(x), y)$ and $\bar{\ell}(f(x), y)$ are $\rho^+$-Lipschitz and $\rho^-$-Lipschitz with respect to $f(x)$ ($0 < \rho^+ < \infty$ and $0 < \rho^- < \infty$) for all $y \in \mathcal{Y}$, respectively, and $w^j$ and $\bar{w}^j$ are both bounded in $[0, \kappa]$. Then, we have

$$\widetilde{\mathfrak{R}}_n(\mathcal{K}_{sp}) \leq \sqrt{2}\kappa c(\rho^+ + \rho^-) \sum_{j=1}^{c} \mathfrak{R}_n(\mathcal{H}_{y_j}),$$

where $\mathcal{H}_y = \{h : x \mapsto f_y(x) | f \in \mathcal{F}\}$ and $\mathfrak{R}_n(\mathcal{H}_y) = \mathbb{E}_{x, \boldsymbol{\sigma}}\left[\sup_{h \in \mathcal{H}_y} \frac{1}{n} \sum_{i=1}^{n} h(x_i)\right]$.

We could obtain the following theorem: Assume the loss function $\ell(f(x), y)$ and $\bar{\ell}(f(x), y)$ are $\rho^+$-Lipschitz and $\rho^-$-Lipschitz with respect to $f(x)$ ($0 < \rho^+ < \infty$ and $0 < \rho^- < \infty$) for all $y \in \mathcal{Y}$ and the loss function $\mathcal{L}_{sp}$ are bounded by $M$, i.e., $M = \sup_{x \in \mathcal{X}, f \in \mathcal{F}, y \in \mathcal{Y}} \mathcal{L}_{sp}(f(x), y)$, with probability at least $1 - \delta$,

$$R(\widehat{f}_{sp}) - R(f^*) \leq 4\sqrt{2}\kappa c(\rho^+ + \rho^-) \sum_{j=1}^{c} \mathfrak{R}_n(\mathcal{H}_y) + M\sqrt{\frac{\log \frac{2}{\delta}}{2n}}.$$

Here, $\widehat{f}_{sp} = \min_{f \in \mathcal{F}} \widehat{R}_{sp}(f)$ and $f^* = \min_{f \in \mathcal{F}} R(f)$ are the empirical risk minimizer and the true risk minimizer, respectively. The proof can be found in the Appendix. This equation shows that $f_{sp}$ would converge to $f^*$ as $n \to \infty$ and $\mathfrak{R}_n(\mathcal{H}_y) \to 0$.

## 4 DISCUSSION AND ANALYSIS

In this section, we focus on why CLIP's methods are effective to echo our motivations. First, we visualize some matrix energies to represent the difficulty of transforming the feature space and the logical label space to the label distribution space (see Figure 3). In this paper matrix energies are computed as described below: First, we set a linear model with bias terms for the transformation of the feature space matrix to the label distribution matrix and the logical label matrix to the label distribution matrix, respectively. The linear model uses the nn.Linear operator in PyTorch 1.8. We used an evaluation platform with a Linux system, a single 3090 RTX GPU shader, the learning rate was uniformly set to 0.002, the batch size was set to throughput all the data at once, and the number of iterations was uniformly set to 1200.

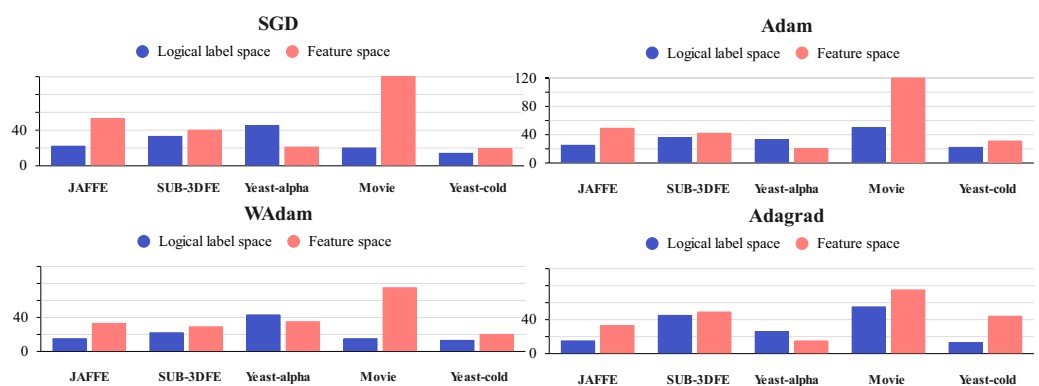

Figure 4: This figure shows the effect of different optimizers on the generation of the energy matrix.

**Maintaining fairness.** The linear models are uniformly overfitted set to ensure fairness in information migration, and they all have a training loss below 0.001.

**Learning rate's effect.** We attempt to obtain matrix energies at multiple learning rates({0.001, 0.01, 0.1}), and we find that the learning rate interferes with the results by less than 3 percent and does not affect the ordinal relationship.

**Optimizer's effect.** In addition, we also considered the effect of optimizers on our method and we evaluated the effect of SGD, Adam, and Adagrad on our experiments. As shown in Figure 4, these four optimizers are essentially homotopic though there are differences in the energy matrices generated by them.

Before the linear model was trained, the transfer matrix energy of the feature space matrix and the transfer matrix energy of the logical label space was essentially equal after using CLIP, with a difference of no more than **2**%. We also tried to perform the training of the linear model on the CPU, due to the limited CPU storage, it is not possible to throughput the entire dataset, which can lead to unfair comparisons. We also perform an exploratory experiment to consider the role of sample weights based on the CLIP2LE algorithm. Specifically, each sample is assigned a learned parameter during training, which is obtained through a 1D convolution and a pooling layer that acts on each sample. We found an overall improvement of **2.4**% in performance across CLIP2LE and will consider the role of sample weights in future work.

In addition, in the Appendix we show the performance of some variants of CLIP.

## 5 LIMITATIONS AND BROADER IMPLICATIONS

Since most of the evaluated datasets are tabular, their feature space dimensions are not uniform. For this, we need to pre-train a CLIP2LE for each dataset, which yields a **great training cost**. This may have led to a small amount of performance degradation on some datasets using our method. Moreover, in this paper, we do not compare with the SOTA method, which is because we focus on the preprocessing ability of CLIP2LE rather than obtaining the best LE results.

Our approach is a pre-processing algorithm that does not involve ethical and moral issues; it is a safe and trustworthy machine-learning algorithm.

## 6 CONCLUSION

In this paper, we propose a generalized label enhancement pre-processing method that can be applied to tasks such as image, text, and speech. Here, we consider label enhancement as a multimodal fusion task, where the issues of fairness and consistency of representations are addressed by a customized CLIP algorithm. In the discussion session, we further illustrate that extant label enhancement algorithms that do not take into account the fairness of the representations result in not all-round performance on multiple benchmarks. Extensive experimental results demonstrate that our proposed CLIP2LE is effective and robust.

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

# A  APPENDIX / SUPPLEMENTAL MATERIAL

This subsection is supplemented with details and data in the 3.1 task.

**Details of the comparison methods.** For the sake of fairness, we utilize the parameter settings recommended in their original works. Specifically, for FCM, we set the parameter $\beta = 2$. For KM, we leverage the Gaussian kernel. For LP, we set the parameter $\alpha = 0.5$. For ML, we set the number of neighbors $k = c + 1$. For GLLE, we select $\lambda$ from $\{0.01, 0.1, ..., 100\}$ and set the number of neighbors $k$ to $c + 1$. For LESC, $\lambda_1$ and $\lambda_2$ are selected from $\{0.0001, 0.1, ..., 10\}$. For LEVI, MLPs with two hidden layers and softplus activation functions are utilized, and the results are reported after 150 training epochs. For LIB, we select $\alpha$ and $\beta$ from $\{0.001, 0.01, ..., 10\}$, and the fully connected networks with 3 layers and sigmoid activation function are leveraged in the proposed method.

**Experimental results.** Experimental results show that our method can effectively improve the performance of existing LE algorithms (see Figure 9 and Figure 11). It is worth noting that the validity of our method is more outstanding in the Cosine metric. Furthermore, we find that consistent with the insights in the main manuscript, our approach improves significantly on non-deep learning algorithms.

| Metric | Canberra ↓ | | | | | | | | Kullback-Leibler ↓ | | | | | | | |
|---|---|---|---|---|---|---|---|---|---|---|---|---|---|---|---|---|
| Method | FCM | KM | LP | ML | GLLE | LESC | LEVI | LIB | FCM | KM | LP | ML | GLLE | LESC | LEVI | LIB |
| Movie | 1.664 | 3.444 | 1.720 | 1.934 | 1.045 | 1.034 | 0.974 | **0.920** | 0.381 | 0.452 | 0.177 | 0.218 | 0.123 | 0.120 | 0.082 | **0.077** |
| SUB-3DFE | 1.020 | 4.121 | 1.245 | 4.001 | 0.820 | 0.799 | 0.637 | **0.611** | 0.094 | 0.603 | 0.105 | 0.565 | 0.069 | 0.064 | 0.042 | **0.041** |
| SJAFFE | 1.081 | 4.010 | 1.064 | 3.138 | 0.781 | 0.561 | 0.600 | **0.531** | 0.107 | 0.558 | 0.077 | 0.391 | 0.050 | 0.029 | 0.032 | **0.027** |
| Yeast-alpha | 2.883 | 11.809 | 4.544 | 11.603 | 1.134 | **0.846** | 1.249 | 0.893 | 0.100 | 0.630 | 0.121 | 0.602 | 0.013 | **0.008** | 0.011 | 0.009 |
| Yeast-cdc | 2.415 | 9.875 | 3.644 | 9.695 | 0.959 | 0.765 | 1,148 | **0.747** | 0.091 | 0.630 | 0.111 | 0.601 | 0.014 | 0.010 | 0.014 | **0.008** |
| Yeast-cold | 0.734 | 2.566 | 0.924 | 2.519 | 0.305 | 0.263 | 0.501 | **0.250** | 0.113 | 0.586 | 0.103 | 0.556 | 0.019 | 0.015 | 0.035 | **0.012** |
| Yeast-diau | 1.895 | 4.261 | 1.748 | 4.180 | 0.671 | **0.480** | 0.689 | 0.621 | 0.159 | 0.538 | 0.127 | 0.509 | 0.027 | **0.017** | 0.023 | 0.022 |
| Yeast-dtt | 0.501 | 2.594 | 0.941 | 2.549 | 0.248 | 0.206 | 0.562 | **0.158** | 0.065 | 0.617 | 0.103 | 0.586 | 0.013 | 0.010 | 0.042 | **0.005** |
| Yeast-elu | 1.689 | 9.110 | 3.381 | 8.949 | 0.902 | 0.727 | 1.093 | **0.670** | 0.059 | 0.617 | 0.109 | 0.589 | 0.013 | 0.009 | 0.014 | **0.008** |
| Yeast-heat | 1.157 | 3.849 | 1.293 | 3.779 | 0.430 | 0.401 | 0.646 | **0.327** | 0.147 | 0.586 | 0.089 | 0.556 | 0.017 | 0.015 | 0.027 | **0.011** |
| Yeast-spo | 0.998 | 3.854 | 1.231 | 3.772 | 0.548 | 0.533 | 0.605 | **0.454** | 0.110 | 0.562 | 0.084 | 0.532 | 0.029 | 0.028 | 0.025 | **0.019** |
| Yeast-spo5 | 0.563 | 1.382 | 0.401 | 1.355 | 0.305 | 0.284 | 0.311 | **0.241** | 0.123 | 0.334 | 0.042 | 0.317 | 0.034 | 0.031 | 0.028 | **0.021** |
| Yeast-sopem | 0.534 | 1.253 | 0.365 | 1.226 | 0.183 | 0.180 | 0.248 | **0.144** | 0.208 | 0.531 | 0.067 | 0.503 | 0.027 | 0.027 | 0.036 | **0.018** |

| Metric | Canberra ↓ | | | | | | | | Kullback-Leibler ↓ | | | | | | | |
|---|---|---|---|---|---|---|---|---|---|---|---|---|---|---|---|---|
| Ours + Method | FCM | KM | LP | ML | GLLE | LESC | LEVI | LIB | FCM | KM | LP | ML | GLLE | LESC | LEVI | LIB |
| Movie | 1.655 | 3.413 | 1.677 | 1.854 | 1.011 | 1.022 | 0.847 | **0.841** | 0.380 | 0.442 | 0.165 | 0.215 | 0.120 | 0.111 | 0.080 | **0.076** |
| SUB-3DFE | 1.012 | 4.022 | 1.115 | 3.985 | 0.765 | 0.791 | 0.630 | **0.610** | 0.083 | 0.533 | 0.095 | 0.411 | 0.063 | 0.061 | 0.040 | **0.040** |
| JAFFE | 1.033 | 3.887 | 1.001 | 3.132 | 0.785 | 0.544 | 0.538 | **0.530** | 0.106 | 0.550 | 0.013 | 0.324 | 0.033 | 0.021 | 0.032 | **0.016** |
| Yeast-alpha | 2.880 | 11.569 | 4.336 | 11.547 | 1.021 | **0.841** | 1.245 | 0.896 | 0.006 | 0.630 | 0.121 | 0.602 | 0.013 | **0.008** | 0.011 | 0.009 |
| Yeast-cdc | 2.410 | 9.863 | 3.643 | 9.610 | 0.921 | 0.753 | 1,145 | **0.740** | 0.087 | 0.611 | 0.110 | 0.599 | 0.013 | 0.010 | 0.012 | **0.007** |
| Yeast-cold | 0.730 | 2.543 | 0.920 | 2.501 | 0.296 | 0.260 | 0.486 | **0.243** | 0.110 | 0.580 | 0.102 | 0.511 | 0.010 | 0.015 | 0.032 | **0.009** |
| Yeast-diau | 1.890 | 4.245 | 1.733 | 4.101 | 0.671 | **0.480** | 0.689 | 0.621 | 0.159 | 0.538 | 0.127 | 0.509 | 0.027 | **0.015** | 0.021 | 0.023 |
| Yeast-dtt | 0.500 | 2.585 | 0.941 | 2.549 | 0.248 | 0.201 | 0.560 | **0.155** | 0.053 | 0.601 | 0.096 | 0.544 | 0.011 | 0.010 | 0.041 | **0.002** |
| Yeast-elu | 1.655 | 9.022 | 3.305 | 8.888 | 0.912 | 0.720 | 1.064 | **0.566** | 0.051 | 0.610 | 0.056 | 0.523 | 0.031 | 0.005 | 0.014 | **0.004** |
| Yeast-heat | 1.123 | 3.489 | 1.113 | 3.536 | 0.225 | 0.400 | 0.619 | **0.223** | 0.140 | 0.513 | 0.066 | 0.433 | 0.012 | 0.013 | 0.028 | **0.011** |
| Yeast-spo | 0.963 | 3.855 | 1.111 | 3.656 | 0.432 | 0.411 | 0.599 | **0.406** | 0.111 | 0.553 | 0.080 | 0.522 | 0.021 | 0.025 | 0.021 | **0.010** |
| Yeast-spo5 | 0.523 | 1.333 | 0.400 | 1.211 | 0.302 | 0.255 | 0.310 | **0.240** | 0.122 | 0.314 | 0.033 | 0.301 | 0.033 | 0.030 | 0.025 | **0.020** |
| Yeast-sopem | 0.532 | 1.233 | 0.313 | 1.225 | 0.180 | 0.153 | 0.212 | **0.140** | 0.202 | 0.523 | 0.060 | 0.477 | 0.020 | 0.021 | 0.035 | **0.015** |

Table 9: Recovery results on 13 real-world datasets. ↓ indicates that "the smaller the better". The following two table blocks represent the 8 LE algorithms using our pre-processing steps. We highlight the best recovery results.

| Metric | Cosine ↑ | | | | | | | | Intersection ↑ | | | | | | | |
|---|---|---|---|---|---|---|---|---|---|---|---|---|---|---|---|---|
| Method | FCM | KM | LP | ML | GLLE | LESC | LEVI | LIB | FCM | KM | LP | ML | GLLE | LESC | LEVI | LIB |
| Movie (Ours) | 0.773 | 0.880 | 0.929 | 0.919 | 0.936 | 0.937 | 0.954 | **0.955** | 0.677 | 0.649 | 0.778 | 0.779 | 0.831 | 0.833 | 0.849 | **0.859** |
| Movie (CCLIP2LE) | 0.775 | 0.856 | 0.925 | 0.912 | 0.934 | 0.937 | 0.965 | **0.967** | 0.675 | 0.644 | 0.772 | 0.770 | 0.811 | 0.815 | 0.810 | **0.866** |
| Movie (TCLIP2LE) | 0.770 | 0.888 | 0.931 | 0.922 | 0.934 | 0.941 | 0.956 | **0.959** | 0.683 | 0.666 | 0.779 | 0.781 | 0.834 | 0.835 | 0.852 | **0.879** |

Table 10: Recovery results on 1 real-world dataset. We highlight the best recovery results.

In this subsection, we discuss the role of other **variants of CLIP** on LE.

In fact, MLP is not necessarily a good choice as a CLIP encoder. Here, we propose two comparison algorithms, one based on CNN (CCLIP2LE) and the other based on Transformer (TCLIP2LE). CCLIP2LE used three-layer convolution and ReLU for the activation function; TCLIP2LE used a three-layer self-attention mechanism and GLU for the activation function. The remaining configurations such as learning rate, optimizer, and batchsize are consistent with the settings of CLIP2LE.

As shown in Table 10, we conducted a comparison experiment on the Movie dataset. We note that TCLIP2LE performs the best, but also has the highest computational cost, and to trade-off speed and

| Metric | Cosine ↑ | | | | | | | | Intersection ↑ | | | | | | | |
|---|---|---|---|---|---|---|---|---|---|---|---|---|---|---|---|---|
| Method | FCM | KM | LP | ML | GLLE | LESC | LEVI | LIB | FCM | KM | LP | ML | GLLE | LESC | LEVI | LIB |
| Movie | 0.773 | 0.880 | 0.929 | 0.919 | 0.936 | 0.937 | 0.954 | **0.955** | 0.677 | 0.649 | 0.778 | 0.779 | 0.831 | 0.833 | 0.849 | **0.859** |
| SUB-3DFE | 0.912 | 0.812 | 0.922 | 0.815 | 0.927 | 0.932 | 0.956 | **0.958** | 0.827 | 0.579 | 0.810 | 0.587 | 0.850 | 0.855 | 0.882 | **0.887** |
| JAFFE | 0.906 | 0.827 | 0.941 | 0.857 | 0.958 | 0.973 | 0.969 | **0.974** | 0.821 | 0.593 | 0.837 | 0.661 | 0.872 | 0.905 | 0.897 | **0.909** |
| Yeast-alpha | 0.922 | 0.751 | 0.911 | 0.756 | 0.987 | **0.992** | 0.989 | **0.992** | 0.844 | 0.532 | 0.774 | 0.537 | 0.938 | **0.953** | 0.932 | 0.951 |
| Yeast-cdc | 0.929 | 0.754 | 0.916 | 0.759 | 0.987 | 0.991 | 0.987 | **0.992** | 0.847 | 0.533 | 0.779 | 0.538 | 0.937 | 0.950 | 0.925 | **0.951** |
| Yeast-cold | 0.922 | 0.779 | 0.925 | 0.784 | 0.982 | 0.986 | 0.970 | **0.988** | 0.833 | 0.559 | 0.794 | 0.565 | 0.924 | 0.935 | 0.881 | **0.938** |
| Yeast-diau | 0.882 | 0.799 | 0.915 | 0.803 | 0.975 | **0.985** | 0.980 | 0.979 | 0.760 | 0.588 | 0.788 | 0.593 | 0.906 | **0.933** | 0.908 | 0.913 |
| Yeast-dtt | 0.959 | 0.759 | 0.921 | 0.763 | 0.988 | 0.991 | 0.965 | **0.995** | 0.894 | 0.541 | 0.786 | 0.546 | 0.939 | 0.949 | 0.866 | **0.961** |
| Yeast-elu | 0.950 | 0.758 | 0.918 | 0.763 | 0.987 | 0.991 | 0.987 | **0.992** | 0.883 | 0.539 | 0.782 | 0.544 | 0.936 | 0.949 | 0.924 | **0.952** |
| Yeast-heat | 0.883 | 0.779 | 0.932 | 0.783 | 0.984 | 0.986 | 0.977 | **0.990** | 0.807 | 0.559 | 0.805 | 0.564 | 0.929 | 0.934 | 0.897 | **0.946** |
| Yeast-spo | 0.909 | 0.800 | 0.939 | 0.803 | 0.974 | 0.975 | 0.978 | **0.982** | 0.836 | 0.575 | 0.819 | 0.580 | 0.909 | 0.912 | 0.903 | **0.925** |
| Yeast-spo5 | 0.922 | 0.882 | 0.969 | 0.884 | 0.971 | 0.974 | 0.979 | **0.983** | 0.838 | 0.724 | 0.886 | 0.727 | 0.901 | 0.908 | 0.909 | **0.924** |
| Yeast-sopem | 0.878 | 0.812 | 0.950 | 0.815 | 0.978 | 0.978 | 0.972 | **0.985** | 0.767 | 0.592 | 0.837 | 0.597 | 0.912 | 0.913 | 0.885 | **0.931** |
| Metric | Cosine ↑ | | | | | | | | Intersection ↑ | | | | | | | |
| Method | FCM | KM | LP | ML | GLLE | LESC | LEVI | LIB | FCM | KM | LP | ML | GLLE | LESC | LEVI | LIB |
| Movie | 0.785 | 0.881 | 0.955 | 0.933 | 0.937 | 0.956 | 0.959 | **0.966** | 0.678 | 0.689 | 0.789 | 0.796 | 0.855 | 0.866 | 0.874 | **0.884** |
| SUB-3DFE | 0.915 | 0.844 | 0.929 | 0.856 | 0.928 | 0.932 | 0.956 | **0.958** | 0.826 | 0.588 | 0.819 | 0.599 | 0.855 | 0.859 | 0.888 | **0.898** |
| JAFFE | 0.910 | 0.855 | 0.946 | 0.869 | 0.973 | 0.971 | 0.979 | **0.985** | 0.829 | 0.603 | 0.856 | 0.669 | 0.883 | 0.913 | 0.899 | **0.926** |
| Yeast-alpha | 0.925 | 0.756 | 0.916 | 0.759 | 0.988 | **0.993** | 0.989 | **0.993** | 0.846 | 0.545 | 0.778 | 0.556 | 0.945 | **0.956** | 0.935 | 0.950 |
| Yeast-cdc | 0.963 | 0.788 | 0.930 | 0.764 | 0.988 | 0.993 | 0.996 | **0.997** | 0.856 | 0.545 | 0.783 | 0.545 | 0.942 | 0.958 | 0.929 | **0.955** |
| Yeast-cold | 0.923 | 0.785 | 0.933 | 0.795 | 0.983 | 0.988 | 0.972 | **0.989** | 0.854 | 0.563 | 0.798 | 0.575 | 0.933 | 0.946 | 0.885 | **0.947** |
| Yeast-diau | 0.899 | 0.803 | 0.918 | 0.811 | 0.986 | **0.986** | 0.982 | 0.981 | 0.765 | 0.563 | 0.789 | 0.594 | 0.912 | **0.945** | 0.912 | 0.919 |
| Yeast-dtt | 0.963 | 0.763 | 0.933 | 0.789 | 0.989 | 0.993 | 0.972 | **0.996** | 0.903 | 0.544 | 0.794 | 0.556 | 0.944 | 0.953 | 0.869 | **0.962** |
| Yeast-elu | 0.955 | 0.762 | 0.935 | 0.763 | 0.987 | 0.992 | 0.989 | **0.992** | 0.889 | 0.545 | 0.789 | 0.563 | 0.945 | 0.953 | 0.933 | **0.959** |
| Yeast-heat | 0.889 | 0.783 | 0.939 | 0.799 | 0.991 | 0.986 | 0.978 | **0.995** | 0.813 | 0.556 | 0.833 | 0.612 | 0.935 | 0.939 | 0.897 | **0.953** |
| Yeast-spo | 0.913 | 0.822 | 0.945 | 0.821 | 0.975 | 0.986 | 0.978 | **0.989** | 0.837 | 0.589 | 0.822 | 0.593 | 0.912 | 0.915 | 0.933 | **0.939** |
| Yeast-spo5 | 0.925 | 0.888 | 0.968 | 0.892 | 0.988 | 0.981 | 0.983 | **0.989** | 0.845 | 0.733 | 0.889 | 0.753 | 0.912 | 0.915 | 0.908 | **0.935** |
| Yeast-sopem | 0.879 | 0.815 | 0.953 | 0.844 | 0.979 | 0.991 | 0.972 | **0.993** | 0.788 | 0.598 | 0.845 | 0.598 | 0.925 | 0.933 | 0.886 | **0.935** |

Table 11: Recovery results on 13 real-world datasets. ↑ means that "the larger the better". The following two table blocks represent the 8 LE algorithms using our pre-processing steps. We highlight the best recovery results.

accuracy, we use CLIP2LE in this paper. Moreover, the only potentially unfair point is the number of iterations; we find that TCLIP2LE converges more slowly, so we set it to have $2\times$ as many iterations as CLIP2LE.

