# OpenReview forum: "CLIP2LE：A  Label Enhancement Fair Representation Method via CLIP"
_ICLR.cc/2025/Conference — Submitted to ICLR 2025_

### Official Review · Reviewer_LyFg · 2024-10-30

**Soundness:** 3
**Presentation:** 3
**Contribution:** 2
**Rating:** 5
**Confidence:** 2

**Summary:**

The paper introduces CLIP2LE, a label enhancement method that leverages CLIP to address challenges in generating label distributions for various learning paradigms, including multi-label, single positive label, and partial-label learning.CLIP2LE proposes a strategy to deal with datasets containing noise and incorrectly labeled samples, A pre-processing algorithm is proposed to ensure fairness in representing the recoded feature and logical labeling spaces.

**Strengths:**

The authors provided reasonable evaluation experiments.The proposed CLIP2LE framework, leveraging CLIP for pre-processing and treating feature and logical label spaces as distinct modalities, is a novel approach.has the potential to improve the performance of existing label enhancement algorithms, which can have a positive impact on various machine learning tasks that rely on accurate label distributions.

**Weaknesses:**

1. Need to pre-train a CLIP2LE for each dataset, this evokes skepticism regarding its practical application. As a preprocessing operation, it is still too heavy.

2. While the paper acknowledges that most evaluated datasets are tabular and have non-uniform feature space dimensions, leading to increased training costs and potential performance degradation, the discussion on this issue is rather cursory.

3. The graphical representation and descriptive exposition of the framework are ambiguous and necessitate greater clarity。

4. There are some tables and expression errors in the experimental results section.

5. the differences in results between different CLIP versions (MaskCLIP and MetaCLIP) are not explained in sufficient detail.

**Questions:**

1. Need to pre-train a CLIP2LE for each dataset, this evokes skepticism regarding its practical application. My interest lies in the subsequent optimization strategies and ensuring its generalizability.

2. The paper acknowledges that non-uniform feature space dimensions of tabular datasets can cause performance degradation and increased training costs. Could you provide more details on the extent of this impact? For example, have you observed any patterns or trends in performance degradation across different datasets with varying degrees of non-uniformity? Also, what steps have been considered to mitigate this issue further?

3. Regarding the differences in CLIP versions, since the text encoder is not used, does the clip type and pure vit visual encoder have an impact on the results? Is there any analysis you can provide?

4. There are some issues with the framework diagram. Each input consists of only one case, so how does contrastive learning work? Is the comparison done within the case? F1 and F2 should represent two different cases. If contrastive learning is used, the introduction of variables is also ambiguous.

5. What is the motivation for Construction of the prototype matrix, why combine these operations, I need a suitable explanation, do not let people think that it is just a random combination to find the right one.

6. Visualization of label distributions recovered by different methods should be provided for intuitive comparison.

7. Table 7 have two "Method without our method".

---

### Official Review · Reviewer_223d · 2024-11-02

**Soundness:** 3
**Presentation:** 3
**Contribution:** 3
**Rating:** 6
**Confidence:** 2

**Summary:**

In this paper, the authors consider that differential contribution of feature space and logical label space, as well as the noise and incorrect labels, derives inconsistent performance of the same label enhancement algorithm on different datasets. To address this, they propose to treat the feature space and the logical label space as two distinct modalities, and adopt contrastive language-image pre-training (CLIP) strategy to recode these modalities as a pre-processing method for the label enhancement algorithm, thus achieving a fair and robust representation. Extensive experiments on multiple benchmarks demonstrate the superiority of the proposed pre-processing algorithm, and reasonable ablation studies confirm the importance of its core designs.

**Strengths:**

- This paper is generally well organized and written.
- The idea of using contrastive language-image pre-training strategy to recode the feature space and the logical label space is well motivated.
- The comparative experiments and ablation studies are comprehensive and offer compelling evidence for the effectiveness of the proposed CLIP2LE framework.

**Weaknesses:**

In my opinion, a core element that makes the CLIP2LE framework effective is treating the logical label vector as tabular data and using the large model TableGPT to encode it. However, it is well-known that large models require substantial memory and computational resources, which can impact the training and inference efficiency of the CLIP2LE framework. The authors should provide some discussion and explanation on this aspect.

**Questions:**

The success of CLIP in visual-language tasks relies heavily on a large-scale dataset containing 400 million image-text pairs. Therefore, does the use of the CLIP strategy for re-encoding feature space and logical label space for label enhancement algorithms also rely on a larger amount of data?

---

### Official Review · Reviewer_jNBk · 2024-11-03

**Soundness:** 2
**Presentation:** 2
**Contribution:** 2
**Rating:** 5
**Confidence:** 3

**Summary:**

The paper presents CLIP2LE, a label enhancement method that integrates feature and logical label spaces using CLIP to improve label distribution quality, particularly addressing issues of dataset noise and annotation inaccuracies. It claims to enhance various learning paradigms through improved label representation.

**Strengths:**

- The use of CLIP as a foundational strategy is innovative in label distribution learning.
- Extensive experimental results demonstrate the effectiveness of CLIP2LE in improving existing label enhancement algorithms across multiple benchmarks.
- The inclusion of formal proofs provides a strong theoretical foundation for the proposed method.

**Weaknesses:**

- I am not familiar with label enhancement, but it appears that the methods discussed in the paper are not widely adopted in multi-label learning. This limits the practical applicability of the proposed improvements, making them less impressive within the context of multi-label learning.

- The evaluation of multi-label learning primarily relies on loss metrics, rather than established metrics such as mAP, precision, and recall. This raises concerns about the overall effectiveness of the method in this specific context.

- The writing quality is subpar, particularly in the introduction, which fails to clearly explain key concepts such as the logical label space and the rationale for using CLIP to treat the logical label space as a separate modality. This lack of clarity hinders the reader's understanding of the paper's contributions.

**Questions:**

Please refer to Weaknesses.

---

### Official Review · Reviewer_vefL · 2024-11-04

**Soundness:** 3
**Presentation:** 3
**Contribution:** 2
**Rating:** 3
**Confidence:** 4

**Summary:**

The manuscript presents the CLIP2LE pre-processing method to address the issues of varying data contributions across datasets and the presence of noise and incorrect labels in the context of Label Enhancement (LE) tasks. Specifically, CLIP2LE aims to approximate the encoded distributions of both feature space and logical label space to a uniform distribution, and it introduces a novel regularization approach to ensure that the semantic information of new features remains intact.

**Strengths:**

The manuscript is well-written and good organized, effectively illustrating the motivation of this paper.

**Weaknesses:**

1. The authors are encouraged to improve the interpretability of their method within the paper. Specifically, it would be beneficial to elucidate which components of CLIP2LE contribute to the robustness against noise and incorrect labels in the dataset.  Furthermore, the manuscript is encouraged to elucidate how TableGPT achieves high-quality representations that ameliorate the sparsity of logical label vectors, and how this affects the treatment of logical label vectors as modalities.
2. It is an intriguing point that warrants discussion: how the varying values obtained from the uniform distributions influenced by the CLIP2LE method might differentially affect the outcomes of Label Enhancement (LE) tasks.
3. The current tables effectively demonstrate the results with and without the incorporation of the CLIP2LE pre-processing method. To further augment the visual representation of the CLIP2LE method's efficacy, it is suggested that the authors consider including a table that explicitly outlines the numerical improvements in performance metrics.
4. In the APPENDIX / SUPPLEMENTAL MATERIAL, the authors note that "the validity of our method is more outstanding in the Cosine metric." It would be beneficial for the authors to provide an explanation for this observation, specifically addressing why the performance of their method differs across various metrics, particularly in comparison to the Cosine metric.

**Questions:**

1. The authors are encouraged to improve the interpretability of their method within the paper. Specifically, it would be beneficial to elucidate which components of CLIP2LE contribute to the robustness against noise and incorrect labels in the dataset.  Furthermore, the manuscript is encouraged to elucidate how TableGPT achieves high-quality representations that ameliorate the sparsity of logical label vectors, and how this affects the treatment of logical label vectors as modalities.
2. It is an intriguing point that warrants discussion: how the varying values obtained from the uniform distributions influenced by the CLIP2LE method might differentially affect the outcomes of Label Enhancement (LE) tasks.
3. The current tables effectively demonstrate the results with and without the incorporation of the CLIP2LE pre-processing method. To further augment the visual representation of the CLIP2LE method's efficacy, it is suggested that the authors consider including a table that explicitly outlines the numerical improvements in performance metrics.
4. In the APPENDIX / SUPPLEMENTAL MATERIAL, the authors note that "the validity of our method is more outstanding in the Cosine metric." It would be beneficial for the authors to provide an explanation for this observation, specifically addressing why the performance of their method differs across various metrics, particularly in comparison to the Cosine metric.

---

### Meta-Review · Area_Chair_Zbu9 · 2024-12-19

**Metareview:**

The paper introduces CLIP2LE, a label enhancement method that integrates feature and logical label spaces through CLIP to enhance label distribution quality. However, the reviewers raised significant concerns regarding the methodology, particularly the substantial memory and computational resource requirements of the proposed approach. The authors failed to provide effective responses to these questions, leaving these critical issues unresolved. Therefore, a rejection is recommended.

**Additional Comments On Reviewer Discussion:**

The authors have not provided any responses to the concerns raised by the reviewers.

---

### Decision · Program_Chairs · 2025-01-22

Reject